# Variation in exposure in neighborhoods of Dhaka, Bangladesh across different environmental pathways: The influence of human behavior on fecal exposure in urban environments

Nuhu Amin[1,2]*, Suraja Raj[3]*, Jamie Green[3], Shahjahan Ali[1,4], Sabrina Haque[5], Yuke Wang[3], Wolfgang Mairinger[3], Tanvir Ahmed[6], George Joseph[7], Mahbubur Rahman[1,8], Christine L. Moe[3]

1 Environmental Health and WASH, Health Systems and Population Studies Division, International Center for Diarrheal Disease Research, Mohakhali, Dhaka, Bangladesh, 2 Institute for Sustainable Futures, University of Technology Sydney, Ultimo, New South Wales, Australia, 3 Center for Global Safe Water, Sanitation, and Hygiene, Emory University, Atlanta, Georgia, United States of America, 4 Department of Epidemiology, Colorado School of Public Health, University of Colorado, Anschutz Medical Campus, Aurora, Colorado, United States of America, 5 Environmental Health Science, Emory University, Atlanta, Georgia, United States of America, 6 International Training Network (ITN), Bangladesh University of Engineering and Technology (BUET), Dhaka, Bangladesh, 7 Water Global Practice, The World Bank, Washington, District of Columbia, United States of America, 8 Global Health and Migration Unit, Department of Women's and Children's Health, Uppsala University, Sweden

* mdnuhu.amin@student.uts.edu (NA); suraja.jeya.raj@emory.edu (SR)

**Editor:** D. Daniel, Gadjah Mada University Faculty of Medicine, Public Health, and Nursing: Universitas Gadjah Mada Fakultas Kedokteran Kesehatan Masyarakat dan Keperawatan, INDONESIA

## Abstract

### Background

Poor sanitation and fecal sludge management contribute to fecal contamination in Dhaka's urban environment. An exposure assessment through ten environmental pathways was conducted using the SaniPath™ tool to understand the exposure to fecal contamination.

### Methods

Data collection took place from 25/04/2017 to 30/01/2018 in ten neighborhoods: four low-income, four high-income, and two transient/floating neighborhoods. A total of 1000 environmental samples were analyzed using the IDEXX QuantiTray/2000 method with IDEXX-Colilert-24® media for the most probable number (MPN) of *E. coli*. Additionally, 823 household surveys, 28 community surveys, and 35 school surveys on exposure-related behaviors were conducted. Bayesian methods were used to estimate monthly *E. coli* exposure levels and population exposure percentages for each environmental pathway in the neighborhoods.

**Data availability statement:** The SaniPath global data is available through request for download at https://www.sanipath.net/data. Study-specific data is available at https://doi.org/10.6084/m9.figshare.30595706.

**Funding:** The study was financially supported by the Bill & Melinda Gates Foundation (grant no. 00010161) through the Rollins School of Public Health at Emory University. icddr,b acknowledges with gratitude the commitment of the Bill & Melinda Gates Foundation and Rollins School of Public Health, Emory University to its research efforts. We acknowledge the World Bank Bangladesh Country Office team for their efforts in partnering, workshops, and contributing to sampling/study design. icddr,b is also grateful to the Governments of Bangladesh, and Canada for providing core/unrestricted support. The funders had no role in study design, data collection and analysis, decision to publish, or preparation of the manuscript.

**Competing interests:** The authors have declared that no competing interests exist.

## Results

Findings revealed that children (aged 5–12 years) in low-income and floating neighborhoods had more frequent contact with most environmental pathways (at least one contact per week or month), except surface water, compared to children in high-income neighborhoods. Dominant exposure pathways varied by neighborhood and socioeconomic status. Children experienced higher estimated monthly fecal exposure doses than adults, primarily through ingestion of contaminated municipal water (all community average = 59.2% of population exposed, either adults or children) and contact with open drains (52.3%) and surface waters (29.0%). Adults were mainly exposed via contaminated municipal water (54.2%), produce (87.0%), and street food (64.5%), which were frequently consumed.

## Conclusions

These results highlight substantial risks of fecal exposure across diverse urban settings in Dhaka. Integrated, multisectoral, and sustainable approaches are critical to reduce exposure and protect public health. Behavior change interventions targeting children and caregivers can further mitigate these risks and help ensure long-term public health improvements.

## Introduction

Dhaka, the capital of Bangladesh, is among the largest, densest, and fastest-growing cities in the world [1]. The sanitation system in Dhaka has been challenged by increasingly rapid growth and has been particularly inadequate in the city's low-income neighborhoods [2,3]. Only 22% of the city's population is connected to the sewer system, while the majority of households use on-site sanitation facilities without proper fecal sludge management [4]. According to a World Bank report on fecal sludge management in 12 cities, an estimated 2% of the fecal waste generated in Dhaka is safely managed, with the majority of fecal waste ending up in the residential environment [5]. A recent study of low-income communities in Dhaka revealed that pathogen levels in open drains and canals remain high even after passing through on-site sanitation systems, such as septic tanks and anaerobic baffled reactors (ABRs), which were not designed to operate effectively without secondary treatment, such as a soil leach field. Without this additional treatment, septic tanks alone do little to remove bacterial and viral pathogens [6]. Poor urban sanitation has been linked to environmental fecal contamination that can have detrimental impacts on public health, such as increased diarrhea and cholera [6–8].

Several studies in Dhaka have detected high levels of fecal contamination in municipal water, surface water, soil, produce, and street food [9–14]. Most of these studies have focused on specific single environmental sources of contamination (e.g., drinking water). While studies of multiple sources of contamination have been conducted in rural Bangladesh [15,16], there is a dearth of information from urban Dhaka exploring multiple

sources of fecal contamination and their contribution to population exposure. The SaniPath™ Exposure Assessment Tool has been implemented in several cities in South Asia and Sub-Saharan Africa [17], and includes customized primary data collection on fecal contamination of the residential environment and human contact behavior and automated Bayesian analyses to estimate risk of fecal exposure [18,19]. Overall, the findings suggest that multiple environmental sources contribute to the risk of exposure, with each playing varied roles across populations and settings. Identifying dominant fecal pathways is necessary to design and target interventions that effectively reduce overall risk and improve health outcomes [17,20,21].

Dhaka is a complex environment, and understanding heterogeneity in behaviors and exposures across the city—such as between low- and high-income neighborhoods or areas with varying WASH infrastructure—is essential. The Citywide Inclusive Sanitation (CWIS) initiative highlights the need for inclusive sanitation strategies that reach vulnerable populations and informal settlements along the entire sanitation service chain and that include a diversity of technologies and innovative solutions to respond to the realities of the challenges faced by low- and middle-income countries [22]. In megacities like Dhaka, substantial investments in water and sanitation systems coincide with stark disparities in household income and access to quality sanitation, it is essential to understand heterogeneity in exposures and prioritize interventions and investments based on evidence in order to achieve effective CWIS [23].

Amin et al. (2019) conducted a cross-sectional study to assess the magnitude of fecal contamination in the environment across low-income, high-income, and transient/floating neighborhoods in urban Dhaka by measuring the frequency and concentration of *E. coli* [24]. However, the study did not assess or compare the levels of *E. coli* contamination with human exposure frequency by neighborhood. This highlights the need to integrate environmental contamination data with human exposure assessments to better understand health risks and prioritize interventions. To address these knowledge gaps, this study aimed to: 1) assess the risks of exposure to fecal contamination across nine environmental pathways in ten urban neighborhoods in Dhaka for both adults and children, 2) compare exposure between neighborhoods with varying characteristics, and 3) examine variations in contact behavior across neighborhoods.

## Methods

### Study sites

A stakeholder meeting, which convened the research partners, local and international NGOs involved in WASH, and government officials responsible for WASH decision-making in Dhaka, was held in Dhaka on February 14, 2017 to inform the selection of study neighborhoods and to ensure the study's relevancy to local WASH and health sector programming. Neighborhoods were selected based on socioeconomic status (low-income, high-income), housing conditions and stability (structured slum vs. unstructured slums, floating/transient vs. permanent populations), WASH infrastructure and services (unimproved WASH vs. improved WASH), and geographic location (Dhaka North City Corporation [DNCC], Dhaka South City Corporation [DSCC]) to help understand variation in risks of fecal exposure across neighborhood types (Table 1). Definitions of the neighborhood categories are provided in S1 Table. We utilised a modified JMP definition of improved and unimproved WASH conditions [25]. Unimproved WASH refers to the absence or inadequacy of safe water on premises (as seen in floating or low-income communities), lack of sanitation with proper fecal sludge management, and limited hygiene-often characterised by open drains and shared or non-functional latrines. In contrast, improved WASH is defined as conditions where communities have piped water supply on premises and toilets connected to a sewer or covered drainage network (i.e., high-income communities). Further detail on the neighborhood selection process and characteristics can be found in the previous publication by Amin et al. (2019).

### Data collection

From 25/04/2017 to 30/07/2017 (pre-monsoon to monsoon season), the SaniPath Exposure Assessment Tool [18,19] was used to conduct multi-pathway exposure assessment in the ten selected neighborhoods in Dhaka. Two separate teams of

**Table 1. Overview of behavioral surveys conducted in households, communities, and schools, Dhaka 2017.**

| Neighborhoods | Household n | Community # survey groups (#total respondents) | Schools #survey groups (# total respondents) |
|---|---|---|---|
| *Floating/transient* | | | |
| Gabtoli | 100 | 5 (84) | 3 (50) |
| Kamalapur | 100 | 5 (88) | 4 (69) |
| *Low-income* | | | |
| Kalshi | 100 | 4 (79) | 5 (83) |
| Shampur | 101 | 5 (90) | 3 (53) |
| Badda | 100 | 5 (87) | 4 (68) |
| Hazaribagh | 100 | 4 (73) | 4 (67) |
| *High-income** | | | |
| Uttarkhan | 42 | – | 4 (77) |
| Motijhil | 33 | – | 4 (75) |
| Gulshan | 65 | – | 1 (16) |
| Dhanmondi | 82 | – | 3 (39) |
| **Total** | **823** | **28 (501)** | **35 (597)** |

* The number of surveys conducted in high-income areas was lower due to challenges in accessing and recruiting participants in these neighborhoods.

- Not collected.

trained field staff conducted the data collection activities. Key informant interviews, mapping and environmental samples were collected by a team of fieldworkers from International Center for Diarrheal Disease Research, Bangladesh (icddr,b), led by the lead author (NA). Behavioral surveys were conducted by a local NGO, DATA, under the supervision of the Emory University team (SR). All data were collected using mobile phones programmed with ODK (Open Data Kit). The environmental sampling forms were administered in English, while the behavioral survey questionnaires were developed in both English and Bengali to ensure cultural and linguistic appropriateness. Both teams received standardized training conducted jointly by icddr,b and Emory teams in Dhaka. The training covered study objectives, ethical considerations, proper use of ODK tools, sample collection protocols, and quality control procedures to ensure consistency and reliability in data collection across all sites.

Key informant interviews with community leaders or local government officials and transect walks were conducted in each neighborhood to identify relevant environmental pathways of potential fecal exposure and contextualize the assessment [24]. Ten environmental pathways were identified, including municipal drinking water, non-municipal drinking water, bathing water, surface water, drain water, flood water, communal/shared latrine surfaces, raw produce, soil and street food (S2 Table). The pathways are described in further detail by Amin et al. (2019) [24]. A total of 1000 environmental samples (10 samples from each environmental pathway as well as 10 soil samples from each neighborhood) were collected according to SaniPath protocols and as described previously [18,24]. Environmental samples were transported from the field to the laboratory at icddr,b and tested for *E. coli* within six hours of sample collection using the IDEXX Quanti-Tray/2000 method and IDEXX-Colilert-24° media (IDEXX Laboratories, Westbrook, ME) according to manufacturer's guidelines [26]. For quality assurance, we included both a negative control and a field blank. The negative control consisted of PBS or sterile water processed in the lab using the same IDEXX methods, while the field blank involved PBS or sterile water taken to the field and "collected" using the same procedures as actual samples, to verify proper sample collection techniques [18]. Results were reported as the most probable number (MPN) of *E. coli* per sample unit. *E. coli* is

used in the SaniPath Tool as the fecal indicator bacteria due to the availability of simple and widely-used testing methods and to allow comparability to national and international guidelines for water and food safety. The laboratory protocol is available in the SaniPath website [19] and has also been uploaded to the protocol.io [27].

From 25/04/2017 to 30/01/2018, behavioral data for household, community, and school surveys were collected following SaniPath Tool protocols [18,19]. The household surveys were conducted with mothers about their behavior and that of their children aged 5–12 years, while community surveys were carried out with groups of approximately 12 males or females (separated by sex) about their behavior and that of children aged 5–12 years. School surveys were administered to students aged 10–12 years in local primary schools about their behaviors and that of their adult parents or caretakers. These surveys aimed to quantify the frequency of contact with various exposure pathways for adults (>18 years) and children (5–12 years). The SaniPath Tool does not assess exposure among children under five, as they typically spend more time in private spaces with their caregivers. Children aged 5–12 years are considered a sentinel population because they are active, independent, and likely to experience high environmental contact (e.g., soil, drains, water bodies, and food). This age group provides a conservative estimate of community exposure risk, reflecting behaviors characteristic of both children and adults in shared environments [17,20,21].

The SaniPath Tool focuses on exposure to fecal contamination in public, not private, spaces for adults and children ages 5–12. This focus was chosen because public areas: 1) can be influenced by sanitation policies, 2) often drive private contamination, 3) are where most environmental contact occurs in crowded urban settings, and 4) allow for feasible data collection. There was no difference in the content of the survey questionnaire across the three respondent groups. However, the focus of each survey type varied slightly based on the setting. Household surveys were conducted to understand individual-level contact behaviors. Community surveys focused on assessing various components of community-level exposures. School surveys aimed to explore exposure pathways specific to school-age children. No participants were compensated for their time in completing the surveys. All the surveys asked the same questions about frequency of: drinking municipal piped water and other drinking water, consuming street food, eating raw (uncooked) produce, contact with surface water, contact with open drains, contact with flood water, and using communal/shared latrines.

Questions about frequency of drinking water, ingestion of raw produce and street food, bathing, and using a communal/shared latrine were framed on a weekly timescale. Respondents were asked about consumption of municipal drinking water and non-municipal drinking water on a different frequency scale. The options given to respondents were "Every day in the past week", "4 to 6 days in the past week", "3 or less days in the past week", "Never", and "Do not know." The reason for using different time scales is that drinking water is a common behavior that may be difficult to recall the number of times consumed in a week. Questions about frequency of contact with open drains, surface water, and flood water were framed on a monthly timescale as these contacts were likely to occur less often.

While collecting water, we also gathered information on household practices related to water treatment. However, these behavioral data were not included in the exposure modeling analysis. Evidence suggests that intermittent water treatment and produce washing alone, particularly without the use of sanitizing agents, has limited impact on *E. coli* removal, highlighting its potential relevance in exposure assessments [28].

We employed a systematic random sampling method to select households in low-income and floating/transient neighborhoods. The process began with fieldworkers identifying a central point within the community, such as a mosque, temple, or other prominent demarcation within the neighborhood. From this central point, fieldworkers systematically selected every third household that met the predetermined selection criteria. This process continued outward from the central point until the desired sample size was achieved. Recruiting participants for surveys was challenging in the four high-income neighborhoods. Door-to-door canvassing was not possible in high-income communities because many high-income communities have gated apartment complexes that are not easily accessible. Therefore, we were unable to randomly select households for surveys in these neighborhoods. Instead, paper surveys were sent home with students (with teacher permission) through schools in these neighborhoods, and their parents were able to complete the surveys and return them

with their children. No community surveys were conducted in high-income neighborhoods due to difficulties with recruitment in gated communities and the lack of structured community leaders in high-income areas [18].

## Data analysis

The primary objective of this study was to assess exposure to fecal contamination through multiple pathways for adults and children in ten Dhaka neighborhoods and identify dominant pathways. The analyses that are built into the SaniPath Exposure Assessment Tool were utilized to estimate the dose of fecal exposure (mean *E. coli* ingested per month) and the percentage of the population at risk for ingesting *E. coli* through a specific pathway (estimated from data reported in the household, community, and school surveys) to determine the dominant exposure pathway(s) for each neighborhood. By using a common metric for dose, the risk can be compared across different pathways.

The exposure assessment approach focuses on estimating distribution parameters, rather than relying solely on a single point estimate, to capture behavior frequencies and fecal contamination concentrations in the environment. Bayesian methods, implemented through JAGS (Just Another Gibbs Sampler) [29], were utilized to estimate these distribution parameters [18]. Using these estimates, Monte Carlo simulations were conducted to assess exposure to fecal contamination for both adults and children across various pathways, incorporating intake volumes derived from existing literature and formative study data [18,24,30] (Fig 1).

An exposure metric ($E$) was calculated for each pathway as the log-transformed product of dose and the percentage of the popoulation exposed. Pathways are classified as dominant if they have an $E$ value greater than 10 (indicating high risk) or fall within one log unit of the maximum $E$ value observed (e.g., if $E_{max} = 5$, dominant pathways are those with $E \in 4 \leq E \leq 5$). If all pathways have $E$ values below 1, no dominant pathway is identified. In the absence of established exposure thresholds for many environmental pathways, these cutoffs were informed by formative and pilot testing conducted in Accra [18]. We defined exposure risk or risk of exposure" as likelihood of being exposed to fecal contamination given someone's behavior and the *E. coli* contamination measured in a specific environmental compartment (Fig 1).

To assess the relationship between the frequency of contact with various environmental compartments and types of neighborhoods, we stratified neighborhoods according to socio-economic status and WASH infrastructure [low-income (considered unimproved WASH conditions), high-income with unimproved WASH, and high-income with improved WASH]. All statistical analyses were conducted using R (version 3.4.1) [31]. All the data used for Dhaka SaniPath deployment is available at SaniPath data portals [19].

The key assumptions for ingestion amounts from different environmental pathways are that intake volumes and durations of exposure are fixed values rather than distributions due to limited literature on their variation. The SaniPath tool uses age-specific parameters (adults vs. children) and pathway-specific multiplication factors to calculate the volume or percentage consumed per exposure event. For water-related pathways, ingestion is calculated using the formula: (Intake Volume/Intake Time) × Duration of Event, while food and surface contact pathways use fixed percentages of consumption per event. The details of those assumptions can be found in the S2 Appendix of Raj et al. (2020) [18].

## Ethics

The study protocol was reviewed and approved by the human subject research committees at the icddr,b (PR-17027) and at Emory University (IRB00051584). Written informed consent was obtained from all respondents during surveys, and for sample collection we obtained consent from the household head. All data were anonymized prior to analysis.

## Results

### General behavior frequency

A total of 823 household surveys, 28 community surveys, and 35 school surveys were conducted among the ten neighborhoods. Sample sizes for each neighborhood can be found in Table 1. Household behavioral surveys indicated

**Fig 1. SaniPath tool analysis diagram with monte carlo simulation (1000 iteration).**

that children in floating and low-income neighborhoods reported more frequent contact (i.e., at least one contact or ingestion event) with nearly all environmental pathways (bathing water = 99% per week, raw produce, street food, and shared latrines >90% per week, floodwater = 71–91% per month, drain water = 67–81% per month, municipal drinking water = 43–67% per week, and non-municipal drinking water = 33–65% per week), except for surface water (34–43% per month), compared to children living in high-income neighborhoods. In high-income neighborhoods, more children reported weekly contact with or ingestion of bathing water (75%), raw produce (70%), municipal water (66%), and street food (58%) compared to the shared latrines (42%), flood water (18%), and monthly contact with drain water (10%) (Table 2). Results from the community surveys in floating and low-income communities showed similar results for children as what was reported in the household surveys, i.e., a high (>50%) percentage of this population reported contact/ingestion with all pathways, except non-municipal water (34–47% per week) (S3 Table). Similar results were found from school surveys of children. School surveys indicated that children from high-income neighborhoods were likely to come into contact with or consume (54%–94%) most environmental pathways except for shared latrines (42% per week) and surface water (44% per month) (S3 Table). The frequencies of adult behaviors across nine environmental pathways are presented in S4 Table.

**Table 2. Frequency of reported children's (age 5-12 years) behaviors (contact/ingestion) across nine environmental pathways based on household surveys in Dhaka, 2017.**

| Neighborhoods and Behavior Frequency | Shared latrines (Week) n (%) | Drain water (Month) n (%) | Bathing water (Week) n (%) | Municipal drinking water (Week) n (%) | Non-municipal water (Week) n (%) | Surface water (Month) n (%) | Produce (Week) n (%) | Street food (Week) n (%) | Flood water (Month) n (%) |
|---|---|---|---|---|---|---|---|---|---|
| cHousehold surveys, n=825 | | | | | | | | | |
| Floating communities (n=200) | | | | | | | | | |
| > 10 times | 131 (65.5) | 89 (44.5) | 82 (41.0) | 86 (43.0) | 99 (49.5) | 33 (16.5) | 37 (18.5) | 37 (18.5) | 112 (56.0) |
| 6-10 times | 58 (29.0) | 29 (14.5) | 110 (55.0) | 0 | 11 (5.5) | 23 (11.5) | 55 (27.5) | 62 (31.0) | 19 (9.5) |
| 1-5 times | 4 (2.0) | 14 (7.0) | 5 (2.5) | 0 | 19 (9.5) | 12 (6.0) | 92 (46.0) | 84 (42.0) | 19 (9.5) |
| Never | 3 (1.5) | 39 (19.5) | 0 | 109 (54.5) | 45 (22.5) | 105 (52.5) | 12 (6.0) | 10 (5.0) | 23 (11.5) |
| I don't know | 0 | 1 (0.5) | 0 | 0 | 0 | 0 | 0 | 1 (0.5) | 0 |
| Not applicable* | 4 (2.0) | 28 (14.0) | 3 (1.5) | 5 (2.5) | 26 (13.0) | 27 (13.5) | 4 (2.0) | 6 (3.0) | 27 (13.5) |
| Low-income (n=401) | | | | | | | | | |
| > 10 times | 236 (58.9) | 225 (56.1) | 94 (23.4) | 263 (65.6) | 52 (13.0) | 85 (21.2) | 50 (12.5) | 69 (17.2) | 207 (51.6) |
| 6-10 times | 103 (25.7) | 44 (11.0) | 278 (69.3) | 7 (1.7) | 14 (3.5) | 34 (8.5) | 113 (28.2) | 131 (32.7) | 59 (14.7) |
| 1-5 times | 23 (5.7) | 55 (13.7) | 28 (7.0) | 1 (0.2) | 44 (11.0) | 52 (13.0) | 199 (49.6) | 167 (41.6) | 100 (24.9) |
| Never | 38 (9.5) | 68 (17.0) | 0 | 130 (32.4) | 282 (70.3) | 222 (55.4) | 36 (9.0) | 25 (6.2) | 21 (5.2) |
| I don't know | 0 | 6 (1.5) | 0 | 0 | 1 (0.2) | 1 (0.2) | 3 (0.7) | 7 (1.7) | 6 (1.5) |
| Not applicable* | 1 (0.2) | 3 (0.7) | 1 (0.2) | 0 | 8 (2.0) | 7 (1.7) | 0 | 2 (0.5) | 8 (2.0) |
| High-income (n=222) | | | | | | | | | |
| > 10 times | 18 (8.1) | 5 (2.3) | 8 (3.6) | 136 (61.3) | 6 (2.7) | 7 (3.2) | 9 (4.1) | 5 (2.3) | 5 (2.3) |
| 6-10 times | 39 (17.6) | 3 (1.4) | 134 (60.4) | 9 (4.1) | 6 (2.6) | 2 (0.9) | 52 (23.4) | 14 (6.3) | 4 (1.8) |
| 1-5 times | 37 (16.7) | 14 (6.3) | 26 (11.7) | 4 (1.8) | 11 (5.0) | 13 (5.9) | 95 (42.8) | 110 (49.5) | 32 (14.4) |
| Never | 62 (27.9) | 144 (64.9) | 2 (0.9) | 10 (4.5) | 123 (55.4) | 140 (63.1) | 12 (5.4) | 35 (15.8) | 121 (54.5) |
| I don't know | 17 (7.7) | 15 (6.8) | 14 (6.3) | 19 (8.6) | 30 (13.5) | 20 (9.0) | 12 (5.4) | 12 (5.4) | 15 (6.8) |
| Not applicable* | 49 (22.1) | 41 (18.5) | 38 (17.1) | 44 (19.8) | 46 (20.7) | 40 (18.0) | 42 (18.9) | 46 (20.7) | 45 (20.3) |

* Not applicable=Unable to collect during surveys. Specific pathways were defined as "not applicable" for one of the floating communities, and therefore those questions were skipped by enumerators while administering the survey.

## Key dominant pathways in Dhaka

Combining the results of the behavior surveys and *E. coli* concentrations in the environmental samples using Bayesian analyses, we examined the dominant pathways of exposure to fecal contamination using the same metric of $\log_{10}$ *E. coli* MPN ingestion per month. The dominant pathways of fecal exposure varied across neighborhoods. For some neighborhoods, there was one very clear dominant fecal exposure pathway (e.g., Gulshan, Fig 2), and in other neighborhoods, there were multiple important exposure pathways (e.g., Ganderia, Fig 2). Despite these variations, several overall patterns were evident.

## Dhaka North City Corporation (DNCC) vs. Dhaka South City Corporation (DSCC)

First, comparing the dominant pathways in the north (DNCC) to those in the south part of the city (DSCC), we observed that exposure to fecal contamination through municipal water was a dominant pathway in three of the five DSCC neighborhoods compared to one of the five DNCC neighborhoods (S1 Fig). This pattern was seen for both child exposure (Fig 2) and adult exposure (Fig 3).

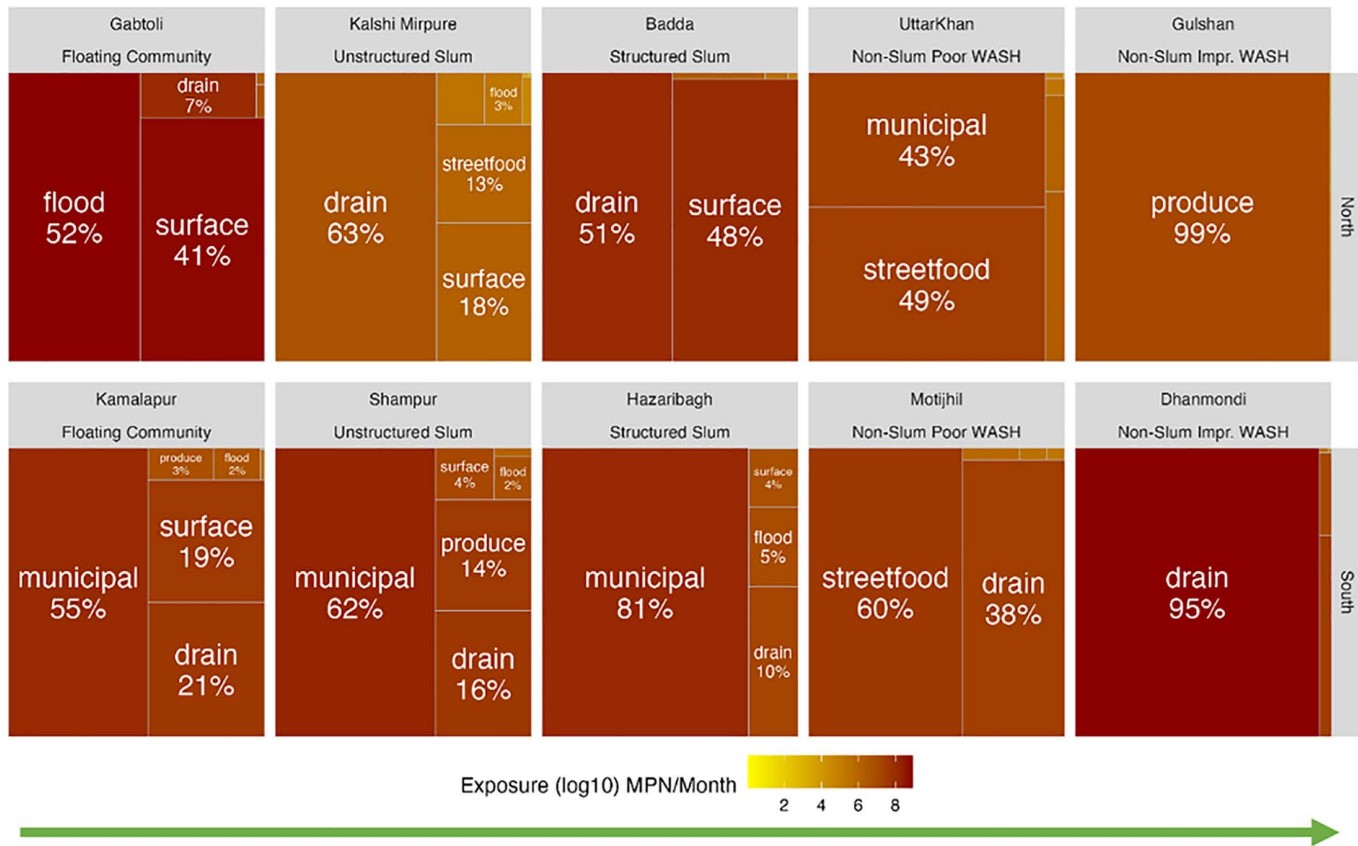

**Fig 2. Total fecal exposure (estimated as log10 MPN *E. coli* ingested per month) for children in ten neighborhoods of Dhaka, 2017.** [The tree plots are arranged left to right according to the neighborhood's socioeconomic status, and the top row is from the North City Corporation while the bottom is from the South City Corporation. The size of the boxes and the darkness of the color reflect the contribution to the total exposure for different pathways within the neighborhood].

### Low-income vs. high-income neighborhoods

We then examined differences in fecal exposure by the socioeconomic status of the neighborhoods. There were two higher income neighborhoods, four slum communities, and two low-income floating communities included in the study. In high-income neighborhoods (Gulshan and Dhanmondi), the dominant exposure pathways were mainly via produce and street food (Fig 3), and the overall fecal exposure was lower (orange shading) for adults in high income neighborhoods compared to low-income neighborhoods (Fig 3). In low-income neighborhoods, there were multiple important fecal exposure pathways including contaminated municipal drinking water, contaminated surface waters and flood waters, as well as exposure to contaminated food (e.g., Gabtoli, Kalshi Mirpure, Baddah, Fig 3). However, in DSCC, exposure to contaminated municipal water eclipsed all the other exposure pathways in low-income neighborhoods (e.g., Kamalapur, Shampur, Hazaribagh, Fig 3). While residents in low-income neighborhoods tended to come into contact with open drains, flood water, and surface water more often, our study showed that the residents in high-income neighborhoods, particularly those with unimproved WASH conditions, still ingested high doses of *E. coli* per month due to elevated concentrations in these pathways (Fig 4).

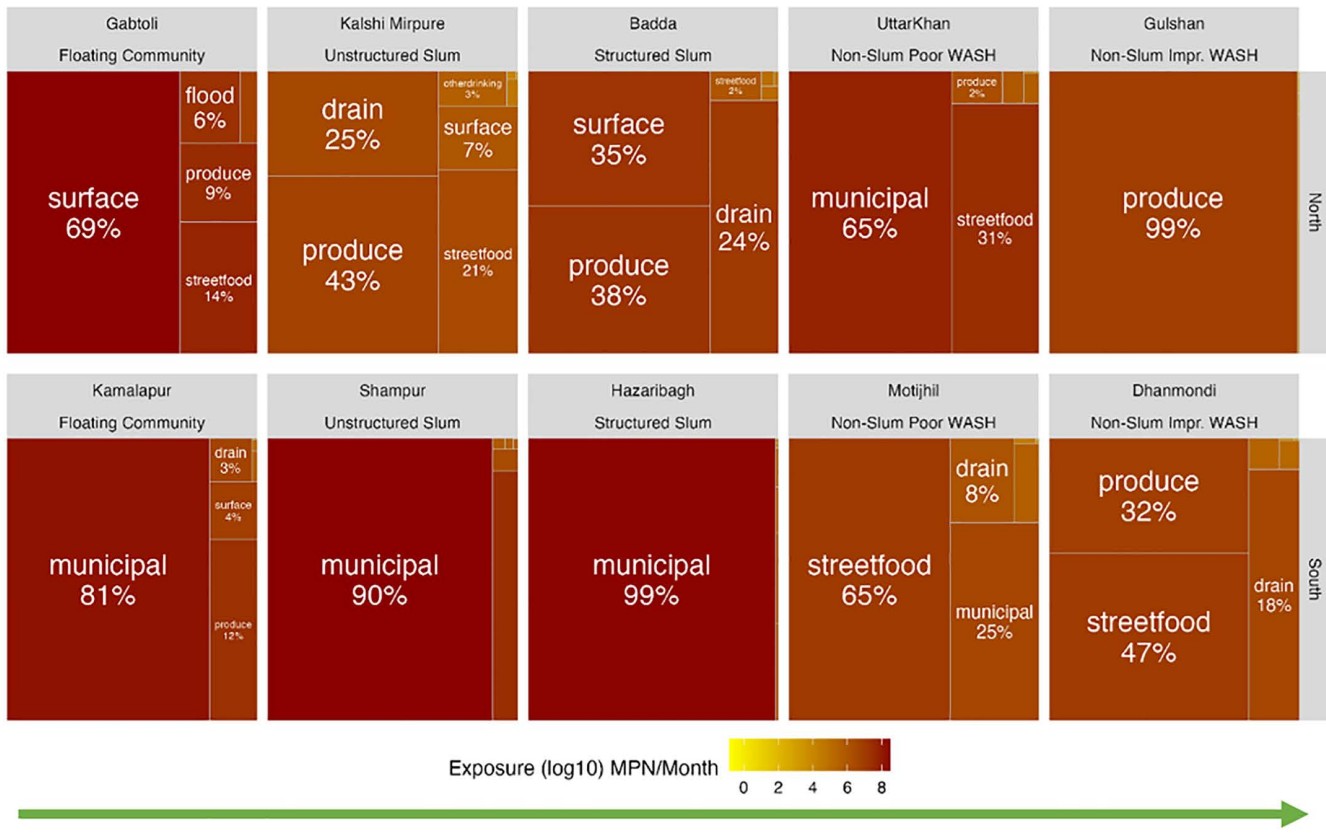

**Fig 3. Total fecal exposure (estimated as log10 MPN _E. coli_ ingested per month) for adults in ten neighborhoods of Dhaka, 2017.** [The tree plots are arranged left to right according to the neighborhood's socioeconomic status, and the top row is from the North City Corporation while the bottom is from the South City Corporation. The size of the boxes and the darkness of the color reflect the contribution to the total exposure for different pathways within the neighborhood.].

### Improved WASH vs. unimproved WASH

Two higher income neighborhoods (Gulshan and Dhanmondi) had improved WASH infrastructure compared to the eight neighborhoods with unimproved WASH facilities. For the neighborhoods with improved WASH, food-related pathways were primarily responsible for fecal exposure (Fig 3). In the neighborhoods with unimproved WASH facilities, exposure to fecal contamination occurred through ingestion of contaminated municipal water and contact with open drains, in addition to fecal exposure through food-related pathways (Figs 2 and 3).

### Child exposure vs. Adult exposure

Differences in behavior between children and adults affected their exposure to fecal contamination in the environment. Children had higher overall fecal exposure compared to adults (predominance of dark red in Fig 2 compared to Fig 3). Children had more exposure to environmental waters, such as open drains, surface water, and flood water, as dominant pathways in both DNCC and DSCC (six neighborhoods) compared to adults. However, exposure to contaminated produce and streetfood in both DNCC and DSCC was a greater risk for adults (Fig 3) compared to children (Fig 2). Pathways that

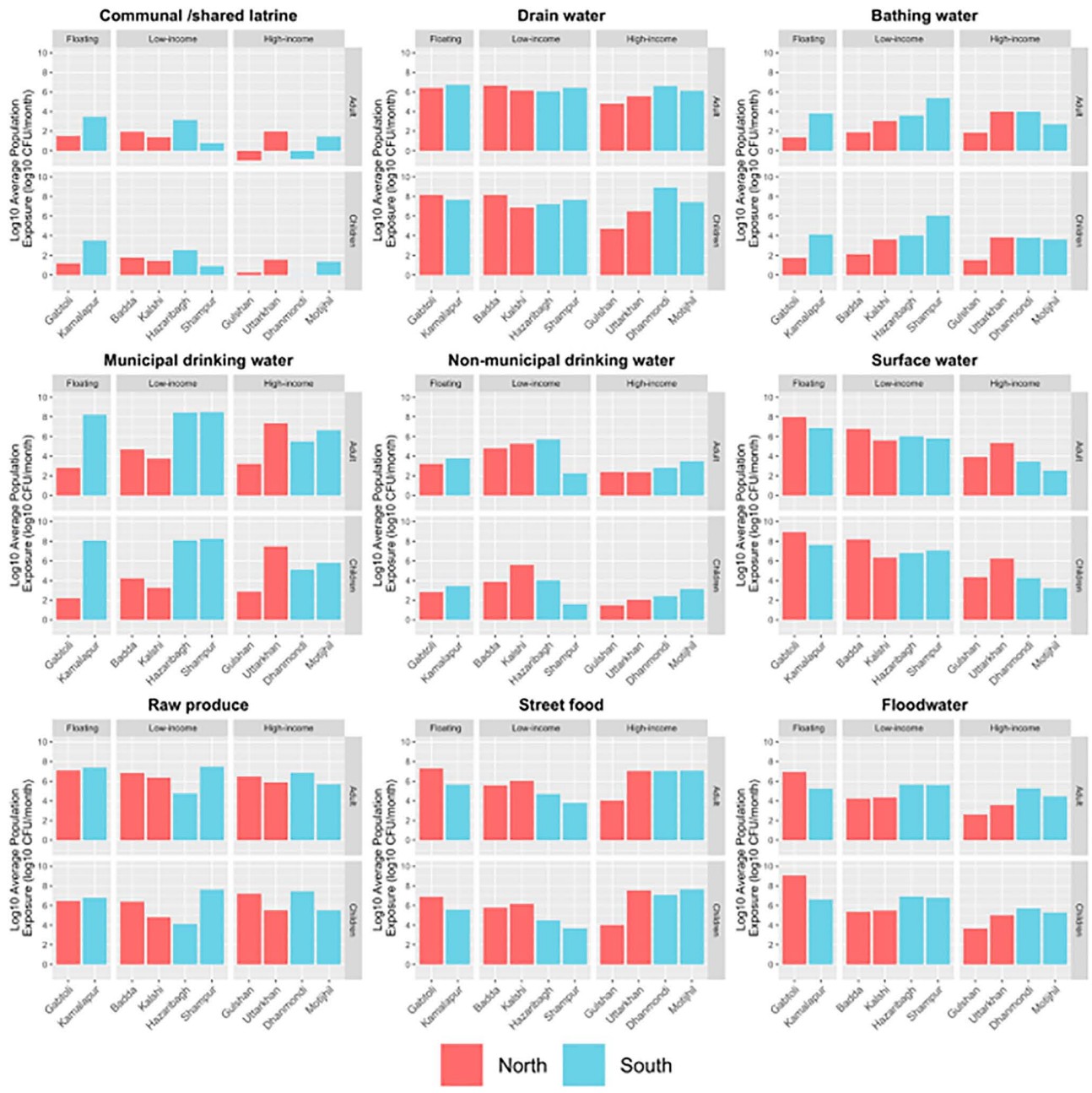

**Fig 4. Average Log10 *E. coli*/month population exposure to fecal contamination through nine different environmental pathways for both adults and children in ten study neighborhoods, Dhaka 2017.** Note: The y-axis for bar charts does not show the negative values for Log$_{10}$ Population Exposure (log$_{10}$ *E. coli* MPN/month) for communal and shared latrines, where estimated exposure was negative on a log$_{10}$ scale.

involved direct ingestion, such as uncooked produce, street food, and municipal water,were the dominant exposure pathways for adults in all neighborhoods and for children in high-income neighborhoods of DNCC (Fig 2).

### Estimated monthly dose of *E. coli*

Overall monthly exposure to *E. coli* ranged from 7.56 to 9.59 $\log_{10}$ MPN/month for adults, with the highest exposure in the Shampur neighborhood, and for children the monthly exposure to *E. coli* ranged from 7.21 to 9.65 $\log_{10}$ MPN/month) with the greatest exposure in the Gabtoli neighborhood. Examining exposure to each specific pathway, it is clear that the greatest risks of fecal exposure were associated with raw produce, street food, drain water, and flood water in most of the study neighborhoods (Fig 4 and S5 and S6 Table). Risks of fecal exposure from municipal drinking water varied by neighborhood with the greatest risks from municipal drinking water for adults in DSCC with very high estimated monthly *E. coli* doses for adults [e.g., Hazaribagh 8.52, Kamalapur 8.23, Motijhil 6.68, and Shampur 8.57 $\log_{10}$ MPN; S5 Table]. Exposure to contaminated surface waters was high in many communities in DSCC and DNCC, depending on their proximity to surface water, and was always higher for children than adults [e.g., Badda 8.35, Gabtoli 9.11, Hazaribagh 7.06, Kalshi 6.58, Kamalapur 8.03, Shampur 7.25, and Uttarkhan 6.49 $\log_{10}$ MPN *E. coli* per month estimated doses for children; S6 Table]. Generally, lower risks of fecal exposure were associated with communal/shared latrines, bathing water (with the exception of the Shampur neighborhood) and non-municipal drinking water (except for some low income neighborhoods).

For children, there was difference in exposure between low-income/floating communities and high-income communities in DNCC for both flood water and surface waters (Fig 4, S7 and S8 Table). There was less variation in exposure to drain water ($10^6$–$10^8$ MPN *E. coli*/month) in all ten neighborhoods, except Gulshan – a high-income neighborhood (Fig 4). In contrast, in low-income/floating neighborhoods of DSCC, municipal water was the overwhelmingly dominant pathway of exposure for children, with monthly doses of *E. coli* exceeding $10^8$ MPN. In low-income neighborhoods in DNCC, exposure was much lower, ranging from $10^2$ to $10^4$ MPN *E. coli*/month (Fig 4). Further examination of the municipal drinking water pathway results indicates that the percentage of children exposed to contaminated municipal drinking water was above 75% in all ten neighborhoods except Gabtoli, where the percent of children exposed to contaminated drinking water was only around 25% (Fig 5).

Examining the dose for children exposed to contamination in municipal drinking water shows that among DNCC neighborhoods, the estimated average dose was approximately $10^3$ MPN *E. coli*/month for all neighborhoods except Uttarkhan ($10^7$ MPN *E. coli*/month). However, in DSCC, the estimated average dose of *E. coli* ingested by children through municipal drinking water ranged from $10^5$ to $10^8$ MPN *E. coli*/month, with greater variation within neighborhoods (Fig 5 and S6 Table).

## Discussion

The SaniPath Exposure Assessment Tool was implemented in ten Dhaka neighborhoods with diverse income levels, WASH conditions, and geographic locations. The aim was to identify key environmental pathways contributing to fecal contamination exposure among children and adults. Results from this exposure assessment indicate widespread fecal contamination in the Dhaka environment. Both children and adults across the ten study neighborhoods in Dhaka came into contact with and ingested high levels of fecal contamination on a monthly basis from various environmental pathways, putting them at risk of infection with enteric pathogens [15]. Reducing exposure by addressing the dominant pathway(s) may substantially reduce the total exposure to fecal contamination in the environment. Certain pathways emerge when considering the most common dominant exposure pathways among all study neighborhoods in Dhaka. These include municipal drinking water, open drains, surface water, flood water, raw produce, and street food (Fig 2 and 3). Our findings are consistent with other research conducted in Dhaka and other parts of Bangladesh on fecal contamination in the environment that reported high levels of *E. coli* detected in various water sources as well as in produce and street-vended

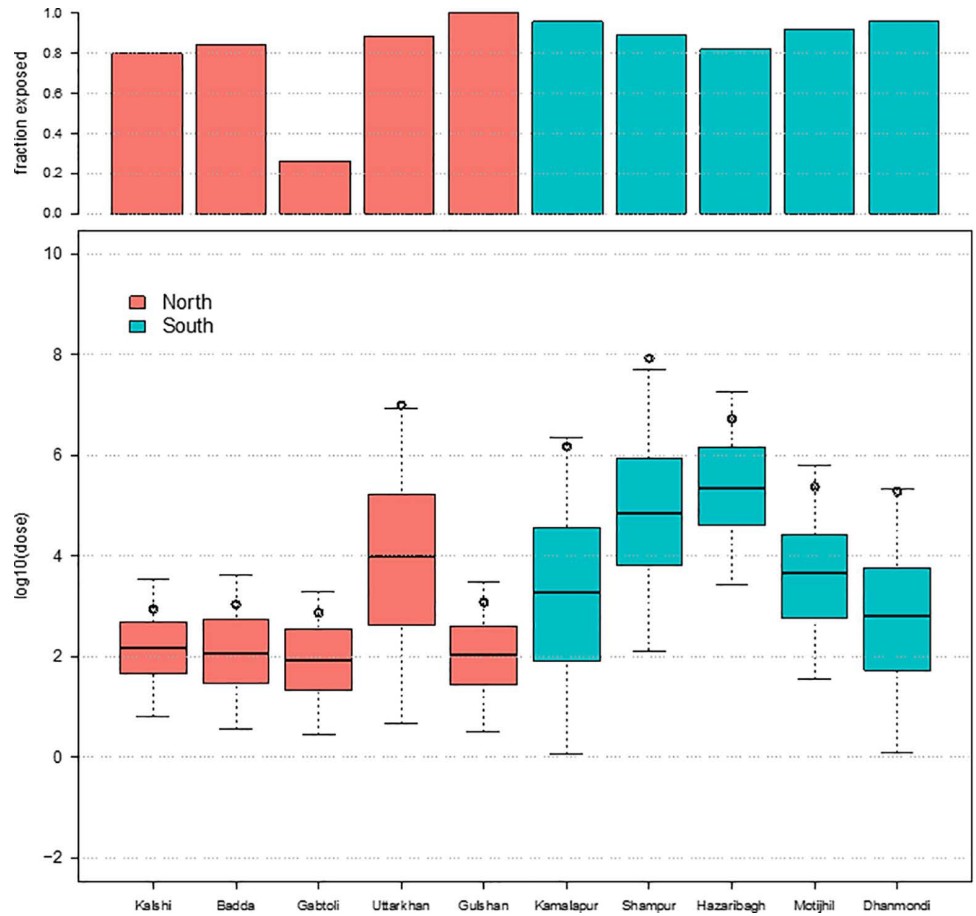

**Fig 5. The proportion of children exposed to fecal contamination and log 10 dose (avg $\log_{10}$ *E. coli* MPN ingested/month) for municipal drinking water across all ten-study neighborhoods.**

foods [4,5,11,12,32,33]. The dominant exposure pathways varied across neighborhoods in Dhaka. However, slums within the same geographic region (north vs. south) revealed similar patterns in dominant pathways, likely driven by comparable sanitation infrastructure. In contrast, non-slum neighborhoods within the same regions showed distinct dominant pathways with no clear geographic clustering pattern. Our study builds on this previous work but adds critical information on frequencies of exposure to multiple environmental pathways that allows us to estimate total monthly fecal exposure from all pathways as well as compare the risks from different pathways and across different neighborhoods using a common metric – MPN of *E. coli* ingested.

The results of this assessment also suggest that the income level and WASH conditions of a neighborhood may influence the frequency of contact with some pathways (open drains, surface water, flood water, and municipal water), but these neighborhood characteristics do not appear to influence contact frequency and exposure from other pathways (i.e., raw produce and street food). While the unequal distribution of WASH services in Dhaka has been well documented, to our knowledge, assessments comparing exposure to fecal contamination between low and high-income urban communities have not been conducted.

## Municipal water

Our results indicate that a high percentage of adults and children were exposed to contaminated municipal water, and higher concentrations of *E. coli* were detected in municipal water samples from DSCC, particularly from Hazaribagh (a low-income neighborhood) compared to DNCC [24]. As discussed by Amin, et al. (2019), potential reasons for the higher water contamination levels in the southern portion of the city include older infrastructure, lack of adequate drainage system, high population density, older, inadequate sanitation facilities, and proximity to the polluted Buriganga river. In Dhaka, municipal water, supplied by Dhaka Water Supply and Sewerage Authority (DWASA), serves as the primary source of drinking and bathing water for an estimated 10.2 million residents. This makes its quality critical to public health, addressing contamination in this system is critical to reducing fecal exposure. Indeed, our study showed that 86% of adults in high-income neighborhoods with improved WASH reported drinking municipal water every day in the prior week (compared to 34% in low-income and 24% in high-income neighborhoods with unimproved WASH). The Dhaka municipal water distribution system faces major challenges, including power outages that reduce pipe pressure and allow contaminants to infiltrate from the surrounding environment. Additional issues include leaking pipes, illegal connections, over-reliance on groundwater extraction, uncontrolled water demand, and the lack of effective metering systems [34,35]. Despite the complexity of these two municipal water distribution systems, this may be the most feasible pathway to target interventions to reduce exposure to fecal contamination because these systems are under direct municipal government control [34] in contrast to food supplies and sanitation systems.

## Raw produce and street food

Our results indicated that the levels of fecal contamination in raw produce were high and eating raw produce and street-vended foods was common in most of the study neighborhoods with little variation between neighborhoods with different income levels, WASH conditions, or geographic location. Fecal contamination of produce and street food in Bangladesh and other LMICs has been reported in numerous SaniPath deployments [21] and studies by others [13,36,37]. It is well documented that the consumption of contaminated foods, which are raw or undercooked, can lead to the spread of food-borne pathogens and cause gastrointestinal infections and disease [13,38,39]. However, food supplies are inherently complex and involve a wide range of sources, stakeholders, and regulatory agencies from "farm to fork", making this exposure pathway challenging to target for intervention.

Consumption of raw produce is common in Bangladesh, where many dishes include ingredients such as raw herbs, vegetables, and spiced water [40]. Contamination of produce may occur at various points along the supply chain from farm to market, and other studies in Bangladesh have reported varying levels of fecal contamination at different points along the chain due to wastewater irrigation, application of fecal material as fertilizer, and unhygienic handling and transport practices [13,41]. While there are risk-mitigating behaviors that can be practiced by farmers, market merchants, and consumers, such as the use of clean water for irrigation, handwashing before and during food preparation, cleaning food preparation surfaces and utensils, washing produce before consumption, these behaviors are not always practiced or are not always effective at eliminating the exposure to produce contamination [40,42].

Street foods are widely consumed across income levels, age groups, and geographic areas in Dhaka and provide convenient and affordable options when people are outside of the home. With the urbanization of Bangladesh continuing to rise, the demand for street food in Dhaka is expected to grow. Consumer surveys have shown that many people report street foods as a major part of their diet and that diverse populations regularly eat street foods, including students, children, wage laborers, housewives, and office workers [38]. With such pervasive consumption of street foods, it is especially concerning that over 90% of street food samples collected during this assessment were contaminated with *E. coli* [24]. Contamination of street foods in Dhaka generally occurs due to poor personal and food hygiene practices among food vendors [43].

There have been significant efforts to improve the safety of street foods in Dhaka. The UN's Food and Agriculture Organization (FAO) partnered with the Dhaka North City and South City Corporations (DNCC), and the government of Netherlands to implement the "Improving Food Safety in Bangladesh" program. Through this initiative, participating food vendors are licensed and provided with a new identifiable food cart, receive training on proper hygiene practices, and their food quality and practices are closely monitored by trained citizens [44]. While this is a positive step, licensing and training of vendors are not required by city authority, and contamination of street foods remains a substantial public health concern. Beyond the FAO program, the national government of Bangladesh has made strides to address food safety concerns by enacting regulations that address food production, importation, processing, stockpiling, and supplying, and to ensure people's rights to access safe foods [45]. However, this authority has not yet created or enforced regulations specifically aimed at street food vendors to improve food safety.

### Open drains, surface water, and flood water

The concentration of fecal contamination in samples from open drains was similar across all study neighborhoods, while contamination levels of surface water and flood water were substantially higher in low-income neighborhoods compared to high-income neighborhoods likely due to higher population density and relatively poorer fecal sludge management (FSM) practices in low-income areas [24]. The extensive fecal contamination in environmental water samples in Dhaka is likely a consequence of unsafe FSM in the city. Indeed, The World Bank's review of FSM in Dhaka in 2014 indicated that only 2% of fecal sludge in Dhaka was effectively treated while the majority of waste was released into the environment without treatment via abandoned onsite sanitation facilities, unsafe collection and emptying of sludge, ineffective treatment of sewage, and sewer leakages [5].

The behavioral data collected in the assessment shows variation in the contact behavior for all three of these pathways across the neighborhood categorizations, with low-income neighborhood residents more likely to report contact with these environmental compartments than high-income neighborhood residents. Low-income "slum" neighborhoods in Dhaka are exceedingly crowded, with over 95% of households measuring less than 14 square meters and a population density of over 220,000 people per km$^2$ compared to non-slums with just over 29,000 people per km$^2$ [46,47]. Furthermore, slum communities in Dhaka, like those included in this assessment, lack access to clean water and basic sanitation services, have broken drainage systems, and experience frequent flooding and waterlogging [46].

Unsurprisingly, those living in low-income communities come into contact with contaminated open drains and flood water more often than those in high-income neighborhoods. A case study conducted by Ahmed, et al. found that adults in urban slums in Bangladesh engaged in activities that exposed them to surface waters, including bathing, ablution, and water collection, which is consistent with our results [48]. Because these waters have been shown to be highly contaminated, any contact may increase the risks of ingestion of fecal contamination and enteric pathogens. It is therefore important to understand the nature and frequency of such behaviors and how to provide safe alternatives to mitigate this exposure.

### Strengths and limitations

This is the largest multi-pathway fecal exposure assessment of its kind conducted in any city. The study collected adult and child behavioral data and environmental data across ten neighborhoods that represented a diverse range of characteristics such as income levels, slum types, WASH quality and access, and geographic regions. Thus, we assessed how risks of exposure from different environmental pathways compared across populations in Dhaka, particularly between high- and low-income communities.

The study has several limitations as well. The assessment was cross-sectional and therefore did not account for any seasonal variation in exposure. We purposely collected all data during the rainy season in Dhaka to represent a "worst

case scenario," but climate change and annual variation in rainfall could create further variation in exposure within a year. Assessments of exposure during both rainy and dry seasons are recommended to better understand seasonal variation. Also, the contamination data was limited to *E. coli* as a fecal indicator. However, a recent study conducted in similar neighborhoods of Dhaka suggested an association between the magnitude of *E. coli* present and the likelihood of enteric pathogens (especially at high levels of *E. coli*) [6].

The behavioral studies posed a particular challenge to conduct in the four high-income neighborhoods of Dhaka. We were not able to conduct any community surveys in these four neighborhoods because there tended to be less cohesion among these neighborhood residents than in the low-income neighborhoods, and opportunities were not available to schedule group surveys in these communities. Collecting household surveys in these neighborhoods was also a challenge because most households were in gated or secured buildings that were not accessible to the enumerators, whereas enumerators in low-income neighborhoods were able to select households randomly and approach them directly. These challenges, along with inherent differences between the different survey methodologies, may have introduced some bias in the surveys. Future studies in high-income communities in Dhaka should consider employing alternative survey methods, for example, telephone or online surveys, as these approaches may be more appropriate for those communities. Detailed strengths and limitations of The SaniPath Tool can be found in Raj et al. (2020).

Although we collected data on household water treatment practices and whether raw produce was washed before consumption, these behavioral factors were not incorporated into the exposure modeling. This exclusion was intentional, as prior evidence indicates that produce washing, especially in the absence of disinfectants, has limited efficacy in removing *E. coli* and may not significantly mitigate exposure [49,50]. Nonetheless, we recognize that such practices can influence actual exposure levels. We have clarified this methodological decision in the revised Methods section. Future modeling studies should consider integrating these behavioral variables to improve the accuracy and contextual relevance of exposure and risk estimates.

## Conclusions

The SaniPath Exposure Assessment Tool used in this study is a unique approach to measuring exposure to fecal contamination in urban environments, combining data on fecal contamination levels in the environment with data on frequencies of behaviors of children and adults associated with exposure to multiple environmental pathways. This approach indicates which neighborhoods have greater exposure to fecal contamination, which environmental pathways are associated with the greatest fecal exposure, and whether the exposure is driven mainly by magnitude of fecal contamination, behavior frequency, or both. The results from the SaniPath assessment in Dhaka demonstrated that children and adults in neighborhoods of varying income levels and WASH conditions across Dhaka experience high levels of exposure to fecal contamination from several pathways, including contact with contaminated open drains, surface water, and flood water, and ingestion of contaminated municipal drinking water, raw produce, and street food. Our results also suggest that the income level and WASH conditions of a neighborhood may be associated with how frequently residents are exposed to open drains, surface water, and flood water; neighborhood income level may also be associated with fecal contamination levels of surface water and flood water. The *E. coli* contamination level of municipal drinking water appears to be associated with geographic location (DNCC vs. DSCC), and the fecal exposure associated with this pathway is particularly striking because of the large population served by this water supply. Also of public health concern is our finding that produce and street food are highly contaminated and widely consumed across Dhaka. This widespread environmental exposure to fecal contamination in Dhaka may be due to multiple factors, such as unsafe and non-functional sanitation systems, unsafe fecal sludge management practices, frequent flooding, rapid unplanned urbanization, and deteriorating water infrastructure [51]. To address these challenges, city authorities must ensure positive water pressure through uninterrupted electricity and backup systems, repair or replace leaking

and corroded pipes, and install inline booster chlorination in high-risk areas. Upgrading water treatment plants with advanced technologies, improving the structural and hydraulic integrity of water distribution systems, and strengthening water quality monitoring are critical steps. Furthermore, public awareness campaigns and community engagement initiatives should be prioritized to promote safe water practices, timely reporting of leaks, and a more resilient water supply system.

Improving the drainage and sanitation systems in Dhaka, including improved onsite sanitation facilities and fecal sludge management, can substantially reduce flooding and sewage overflow, subsequently reducing fecal contamination of flood water and surface water. This is a considerable undertaking and efforts have already begun to achieve these improvements [23]. To mitigate the environmental exposure to fecal contamination and pathogens through flooding, it is important to adopt multisectoral, integrated, inclusive, and long-term sustainable approaches such as appropriate urban development plans synced with the existing national-level policies and plans [24,52], along with appropriate behavior change interventions [53] for children and caregivers to reduce exposure. In addition, urban infrastructure development; innovative, affordable and sustainable technological improvement of the sanitation networks- including latrines, drainage, protection of surface water bodies, and community-based wastewater treatment plants-are critical. Building the skills of municipal water and sanitation engineers and sanitation workers is equally important to enhance overall environmental management and reduce exposure. An integrated approach to sanitation, such as the CWIS model, which combines sanitation improvements with water supply, solid waste management, and drainage systems, could deliver substantial public health benefits for urban populations [54]. Further actions may address the contact behavior with open drains, flood water, and surface water, particularly among residents in low-income areas. For example, covering open drains within pedestrian and residential areas and conducting behavior change campaigns aimed at reducing the use of surface water for household and recreational activities may reduce the frequency that people are exposed to these pathways.

Although Bangladesh declared zero open defecation in 2015, our results suggest that Dhaka city's drains and surface water bodies have high levels of fecal contamination. Recent studies from icddr,b also indicate that most of this contamination comes from the effluent of onsite sanitation systems [6,55,56]. Although open defecation has been eliminated, improper management of latrine effluent and fecal sludge continues to contaminate Dhaka's drains and water bodies. To our knowledge, there are currently no community-based wastewater treatment facilities in Dhaka City. Only 20% of wastewater (e.g., community, hospital, industries, etc.) in Dhaka city is run through a functional treatment plant. The rest of the wastewater carries biological, chemical, and pharmaceutical waste into surface water bodies and other parts of the residential environment. Furthermore, the lone sewage treatment plant of Dhaka WASA is situated in Pagla, and the treatment capacity of that plant is not fully utilized because of limitations in the sewerage network. Therefore, community-level wastewater treatment technology [57] needs to be developed and implemented to reduce biological, chemical, and pharmaceutical contamination of the environment [58]. Improving the sanitation system is essential to reduce fecal contamination of drain and surface water and directly impacts the quality of municipal drinking water [59,60], raw produce [13], and street food [36].

Additional research may be conducted to better understand exposure to fecal contamination and specific pathogens in Dhaka and other urban areas in LMICs. Specifically, research is needed to: 1) better understand the linkages between poor sanitation and food contamination along the "farm-to-fork chain", 2) design and evaluate the effectiveness of interventions aimed at improving food hygiene practices among farmers and food vendors, as well as 3) understand attitudes and practices around contact with surface water and open drains. The SaniPath tool can identify high-risk areas for CWIS interventions by assessing behavioral exposure and contamination data. The Bangladesh Government is currently using Shit Flow Diagrams (SFD) through the national sanitation dashboard [61] as a tool for municipal planning. City authorities can integrate the SaniPath tool into urban planning frameworks to prioritize high-risk areas and design targeted interventions.

## Supporting information

**S1 Table. Definitions of neighborhoods for SaniPath Dhaka deployment, 2017.**
(DOCX)

**S2 Table. Definitions of ten environmental sample types, Dhaka 2017.**
(DOCX)

**S3 Table. Frequency of reported child (age 5–12 years) behaviors (contact/ingestion) across nine environmental pathways based on household, community, and school surveys in Dhaka, Bangladesh (2017).**
(DOCX)

**S4 Table. Frequency of reported adult behaviors (contact/ingestion) across nine environmental pathways based on household, community and school surveys in Dhaka, Bangladesh (2017).**
(DOCX)

**S5 Table. Summary of estimated monthly *E. coli* doses (log10 MPN) for adults by exposure pathway and neighborhood: Dose-Adults.**
(DOCX)

**S6 Table. Summary of estimated monthly *E. coli* doses (log10 MPN) for children by exposure pathway and neighborhood: Dose- Children.**
(DOCX)

**S7 Table. Comparing fecal exposure risks for children (age 5–12 years) by pathway and neighborhood characteristics.**
(DOCX)

**S8 Table. Comparing fecal exposure risks for adults by pathway and neighborhood characteristics.**
(DOCX)

**S1 Fig. The proportion of adults exposed to fecal contamination and log 10 dose (avg $\log_{10}$ *E. coli* MPN ingested/month) for municipal drinking water across all neighborhoods.**
(TIF)

## Acknowledgments

We are grateful for the support and advice of our project officers at the Bill & Melinda Gates Foundation: Radu Ban, Erica Coppel, and Alyse Schrecongost. We are grateful for the contribution of Jimi Michiel and Kyndall White to the development of mobile data collection tools. We thank Casey Siesel for his thoughtful input and review of this paper. The team at the World Bank Bangladesh office, including Rokeya Ahmed and Bakhtiar Sohag, provided valuable insight on neighborhood selection and study design, as well as the convening of stakeholders and workshop planning. This work would not have been possible without the efforts of the SaniPath Dhaka study team at: 1) icddr,b, including laboratory managers, Shimul Das and Arefeen Haider; field managers, Palash Mutsuddi and Mohammad Russel; and the sample collectors, transporters, and data entry staff, and at 2) Data Analysis and Technical Assistance (DATA), including Imrul Hassan, Mohammad Zahidul Hassan, and all enumerators who conducted behavioral surveys. We are also grateful for the assistance of Dr. Leanne Unicomb in recruiting schools in high-income neighborhoods to participate in SaniPath surveys. Finally, we are thankful to the communities of Dhaka who invited us into their neighborhoods, schools, and homes to conduct this research.

## Author contributions

**Conceptualization:** Nuhu Amin, Suraja Raj, Yuke Wang, George Joseph, Mahbubur Rahman, Christine L. Moe.

**Data curation:** Shahjahan Ali, Yuke Wang.

**Formal analysis:** Nuhu Amin, Yuke Wang.

**Funding acquisition:** Suraja Raj, Sabrina Haque, George Joseph, Mahbubur Rahman, Christine L. Moe.

**Investigation:** Nuhu Amin, Shahjahan Ali, Tanvir Ahmed.

**Methodology:** Nuhu Amin, Suraja Raj, Shahjahan Ali, Yuke Wang, Wolfgang Mairinger, George Joseph, Christine L. Moe.

**Project administration:** Nuhu Amin, Suraja Raj, Jamie Green, Shahjahan Ali, Mahbubur Rahman.

**Resources:** Nuhu Amin, Suraja Raj, Jamie Green, Sabrina Haque, Wolfgang Mairinger, George Joseph, Christine L. Moe.

**Software:** Nuhu Amin, Yuke Wang.

**Supervision:** Tanvir Ahmed, George Joseph, Mahbubur Rahman, Christine L. Moe.

**Validation:** Nuhu Amin, Jamie Green, Shahjahan Ali, Sabrina Haque, Yuke Wang, Wolfgang Mairinger, Tanvir Ahmed, Mahbubur Rahman, Christine L. Moe.

**Visualization:** Sabrina Haque, Yuke Wang.

**Writing – original draft:** Nuhu Amin.

**Writing – review & editing:** Suraja Raj, Jamie Green, Shahjahan Ali, Sabrina Haque, Yuke Wang, Wolfgang Mairinger, Tanvir Ahmed, George Joseph, Mahbubur Rahman, Christine L. Moe.

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
