## [Decision Letter · Decision Letter 0]

20 May 2025

Dear Dr. Amin,

Thank you for submitting your manuscript to PLOS ONE. After careful consideration, we feel that it has merit but does not fully meet PLOS ONE’s publication criteria as it currently stands. Therefore, we invite you to submit a revised version of the manuscript that addresses the points raised during the review process.

We look forward to receiving your revised manuscript.

Kind regards,

D. Daniel, Ph.D.

Academic Editor

PLOS ONE

Journal Requirements:

Reviewers' comments:

Reviewer's Responses to Questions

**Comments to the Author**

1. Is the manuscript technically sound, and do the data support the conclusions?

Reviewer #1: Yes

Reviewer #2: Partly

Reviewer #3: Partly

2. Has the statistical analysis been performed appropriately and rigorously?

Reviewer #1: I Don't Know

Reviewer #2: I Don't Know

Reviewer #3: I Don't Know

3. Have the authors made all data underlying the findings in their manuscript fully available?

Reviewer #1: Yes

Reviewer #2: No

Reviewer #3: No

4. Is the manuscript presented in an intelligible fashion and written in standard English?

Reviewer #1: No

Reviewer #2: Yes

Reviewer #3: Yes

Reviewer #1: The authors present an interesting study characterizing fecal pollution sources, pathways, and exposures across communities in Dhaka. Please expand on the methods and revise the conclusion. More information regarding the SaniPath Tool and its incorporation into the study is needed as well.

Line 22: nine environmental pathways, not nine-environmental pathways

Line 23: Does the SaniPath tool need at trademark?

Line 66: Please expand a bit on the SaniPath tool

Line 93: Can you list the date of this meeting?

Data collection: Did you have any controls for the microbial testing?

Line 164: Please remove "like the mean".

Define JAGS

Line 169: Please include references for the "existing literature and formative study data". Is there a table describing this data?

You also repeat the exposure assessment methodology in lines 169-171. Please remove.

The second half of the paragraph is repeated. Please revise.

Data analysis: additional information is needed here. Are there any equations for dose? More information on the software is needed too.

Survey design and questions need to be included in methodology. There is some information in the footnotes under Table 2, but this needs to be included in the methods section.

The Results section has several typos and the figures are very blurry and hard to review. I had trouble accurately assessing the results given the quality of the figures.

Line 412: Please include references for "and studies by others".

Line 591: Is there a typo for SFD.

The conclusion is very long and much of the information discussed should be included in the discussion. Please revise.

Reviewer #2: Overall comments:

The authors investigated fecal exposure from different routes in urban Dhaka using a SaniPath methodology. There are some places where the manuscript could be strengthened, outlined in detail below.

Specific comments:

Abstract

It’s unclear to me what the %’s reported in the results section of the abstract mean. Please provide a more detailed description of this.

Introduction

I’m not entirely clear what the contribution of this paper is compared to a similar paper that was previously published by the same authors from this data (Amin et al. 2019). There seems to be a fair amount of overlap in the methods and discussion sections, with both papers presenting relatively similar take-aways. It would be helpful if the specific results from that paper were outlined in the introduction to explain the specific knowledge gap left by that paper that this paper aims to fill.

Methods

The abstract says that 900 environmental samples were collected and analyzed for E.coli whereas the methods sections instead says that 1000 environmental samples were collected and analyzed. Please clarify this discrepancy.

The methods specify that data was collected to understand and compare exposure between adults (>18 years old) and children (5–12 years old). Please clarify and provide justification in the manuscript as to why children <5 years old were not included in this study. Children <5 are typically considered an especially vulnerable population for health concerns related to fecal contamination exposure.

Lines 169-176 seem to be a repeat of what you said in lines 163-169 in slightly different words. Please review and rectify.

Were questions asked about whether the respondent did anything to treat the water before drinking it? Or whether they cleaned the raw produce before eating it? These types of practices could change exposure, so it would be helpful to clarify these points.

What assumptions were used for calculating the amount ingested from contact with things like drinking water, surface water, flood water, etc.? These assumptions play a key role in estimating and comparing exposure among the different pathways that underlie your results and conclusions, and it’s not clear to me from the methods section what data was used to estimate ingestion/exposure based on reported frequency of contact with each.

Please update the methods section to include more information about the survey conducted, such as who performed the survey, language of survey, and relevant training of enumerators. Details of who collected the environment sample should also be included.

Results

At the beginning of the results section (e.g., lines 197-212 such as “Household behavioral surveys suggested that children in floating and low-income neighborhoods reported more frequent contact (i.e., at least one contact or ingestion event) with nearly all environmental pathways”), it is not clear to me what the time period for these contacts is – one contact per day, week, etc? Please specify this more clearly throughout.

From Figure 1, the exposure pathways seem relatively neighborhood specific. I suggest providing more discussion of this among the manuscript.

The text on Figure 3 is difficult to read and I would recommend using a higher resolution for this image.

From Figure 3, it seems that many pathways are important sources of high levels of fecal exposure. Given this figure, I’m not entirely convinced about some of the dominant pathway findings between children/adults or northern/southern areas since many pathways seem to lead to high levels of exposure and need to be addressed. For example, lines 294-301 and 303-313 are not entirely convincing based Fig 3. I suggest providing stronger justification for these points, or revising the take-aways. For example, were any statistical tests done to support the conclusions in results about different among different neighborhoods, adults/child, etc?

Lines 350-351: “10310^3103 MPN” is unclear, please rectify.

It’s not clear exactly what poor WASH vs improved WASH means in this context when you present results comparing the two. Can you provide more details about what you mean by both?

Was any information collected about household characteristics that could be added to help interpret and contextualize the findings?

Discussion

Can the estimated values of Ecoli ingested be put into context of diarrhea risk? This would help interpret the findings of logs of Ecoli ingested for different pathways presented in the results section.

Supporting data

The authors mentioned that “All relevant data are within the manuscript and its Supporting Information files.”, but I did not see the underlying data submitted with the SI files.

Reviewer #3: The authors are reporting environmental and behavioral data collected and analyzed via the SaniPath tool to assess human exposure to fecal indicator bacteria across several pathways in Dhaka, Bangladesh. Their work provides data that can inform local risk management strategies to reduce exposure to fecal contamination in Dhaka. Please see below for further comments and suggestions:

My biggest source of confusion is the aims. “Risks of exposure” and “exposure risks” are unclear and uninformative phrases that are inconsistent with traditional microbial risk assessment approaches. Are the authors estimating exposure (e.g., via a dose) or are they estimating a risk (e.g., of a non-cancer health endpoint, like an infection). Similarly, “exposure behavior” is not clear to this reader what is being assessed.

Most of the analysis methods for this paper were completed by the SaniPath Exposure Assessment tool, whose methods were described in another paper. As a result, this paper is not easy to understand without reading the other paper. I suggest the authors BRIEFLY describe the key inputs and data processing steps of the tool.

Although they have provided numerous data tables in the supplement, the authors have not made a cleaned or de-identified dataset available.

Line 129-138: Please explain the difference between household vs. community vs. school surveys. Was there any difference in content between the different versions of the surveys. Were participants compensated for time they gave to complete surveys?

166 - What is JAGS? Please explain/define.

Lines 169-178. More in text description of exposure and risk methodology is encouraged. As is, it is not clear what equations were used to estimate the dose and risk. In addition, if the authors are providing a risk, they should clarify what the outcome measure is (i.e., hazard quotient). Please explain.

Line 315. Could you provide more details about how the doses were estimated? I presume dose was a combination of a level of e.coli and a model that incorporates differences in ingestion rates and other behaviors. Given that the reader does not have access to the underlying inputs of e.coli and behavior, the reader cannot ascertain the extent to which the differences in dose were attributable to the level of e.coli or the behavior. Please provide a sensitivity analysis for thiss.

Line 296. Please describe “improved wash infrastructure”

Line 334 - How was the significant difference determined? If it was determined statistically, these methods need to be described in the manuscript and results of hypothesis testing provided. This comment applies throughout the manuscript when assertions of differences are described.

Line 364. The phrase “risk of exposure” is not clear. Please rephrase and present risk estimates that relate to a health outcome (e.g., risk of infection or symptoms). This would provide more actionable information to inform risk mitigation strategies.

Line 570 - What does icddr,b mean? This is only written out in author affiliations. Needs to be explained at first mention in MS.

**Do you want your identity to be public for this peer review?** For information about this choice, including consent withdrawal, please see our Privacy Policy

Reviewer #1: No

Reviewer #2: No

Reviewer #3: No

---

## [Author Response · Author response to Decision Letter 1]

5 Aug 2025

Response to reviewer's comments:

Dear Reviewers,

Thank you for your valuable comments and suggestions. I have addressed all of them in the revised manuscript. All changes have been made using track changes, and the corresponding line numbers are indicated in the response form.

I hope this revision adequately clarifies all your queries.

Editor’s comments:

Dear Dr. Amin,

Thank you for submitting your manuscript to PLOS ONE. After careful consideration, we feel that it has merit but does not fully meet PLOS ONE’s publication criteria as it currently stands. Therefore, we invite you to submit a revised version of the manuscript that addresses the points raised during the review process.

Response: We have included the protocols in the protocols.io with the DOI: 10.17504/protocols.io.q26g7nbe9lwz/v1.

We have put the information on page 5, lines 141-42: The laboratory protocol is available in the SaniPath website [1] and has also been uploaded to the protocols.io [2].

Journal Requirements:

Response: We have corrected the file naming and formatting

Response: We have checked the funding information, and all information is correct: The study was financially supported by the Bill & Melinda Gates Foundation (grant no. 00010161) to the Rollins School of Public Health at Emory University.

Response: The SaniPath data is available for download at https://www.sanipath.net/data

Reviewers' comments:

Authors responses to Questions

Comments to the Author

1. Is the manuscript technically sound, and do the data support the conclusions?

Reviewer #1: Yes

Reviewer #2: Partly

Reviewer #3: Partly

2. Has the statistical analysis been performed appropriately and rigorously?

Reviewer #1: I Don't Know

Reviewer #2: I Don't Know

Reviewer #3: I Don't Know

3. Have the authors made all data underlying the findings in their manuscript fully available?

Reviewer #1: Yes

Reviewer #2: No

Reviewer #3: No

Response: The data has already been made freely available for download at https://www.sanipath.net/data.

4. Is the manuscript presented in an intelligible fashion and written in standard English?

Reviewer #1: No

Reviewer #2: Yes

Reviewer #3: Yes

Review Comments to the Author

Reviewer #1:

The authors present an interesting study characterizing fecal pollution sources, pathways, and exposures across communities in Dhaka. Please expand on the methods and revise the conclusion. More information regarding the SaniPath Tool and its incorporation into the study is needed as well.

Response: Thank you for your thoughtful comments and suggestions. We agree with your observations and have revised the manuscript to provide additional clarity on the SaniPath Tool, the methods, and the study's conclusions. Specifically, the following changes have been made:

Introduction (Page 2, Lines 70–74):

We have clarified the definition and functioning of the SaniPath Tool, emphasizing its role in assessing environmental fecal exposure through multiple pathways.

Introduction (Page 3, Lines 90–96:

We have explained how this study differs from the previous SaniPath study in Dhaka (Amin et al. 2019):

"Amin et al. (2019) conducted a cross-sectional study to assess the magnitude of fecal contamination in the environment across low-income, high-income, and transient/floating neighborhoods in urban Dhaka by measuring the frequency and concentration of E. coli [24]. However, the study did not assess or compare the levels of E. coli contamination with human exposure frequency by neighborhood. This highlights the need to integrate environmental contamination data with human exposure assessments to better understand health risks and prioritize interventions."

Methods (Page 5, Lines 160–164:

We have added information regarding laboratory quality assurance and quality control (QA/QC), including the use of field blanks, duplicates, and data sources and protocols.

“For quality assurance, we included both a negative control and a field blank. The negative control consisted of PBS or sterile water processed in the lab using the same IDEXX methods, while the field blank involved PBS or sterile water taken to the field and "collected" using the same procedures as actual samples, to verify proper sample collection techniques”

Methods (Page 7, Lines 194–202):

We have expanded the description of the questionnaire used for exposure data collection:

"Questions about frequency of drinking water, ingestion of raw produce and street food, bathing, and using a communal/shared latrine were framed on a weekly timescale. Respondents were asked about consumption of municipal drinking water and non-municipal drinking water on a different frequency scale. The options given to respondents were “Every day in the past week”, “4 to 6 days in the past week”, “3 or less days in the past week”, “Never”, and “Do not know.” The reason for using different time scales is that drinking water is a common behavior that may be difficult to recall the number of times consumed in a week. Questions about frequency of contact with open drains, surface water, and flood water were framed on a monthly timescale as these contacts were likely to occur less often."

Methods (Page 8, Lines 252-258):

We have clarified the steps of the exposure analysis:

"An exposure metric (*E* ) was calculated for each pathway as the log-transformed product of the estimated dose and the percentage of the population exposed. A pathway was classified as dominant if its *E* value was greater than 10 (indicating high risk) or within one log unit of the maximum *E* value observed (e.g., if *E* ₘₐₓ = 5, dominant pathways are those with *E* ∈ 4 ≤E≤5). If all *E* values were below 1, no dominant pathway was identified. In the absence of established thresholds for environmental pathways, these cutoffs were informed by formative and pilot testing conducted in Accra [18]."

We have also added website links within the citations and revised the conclusion section to better reflect these methodological clarifications and highlight the integration of exposure behavior and E. coli contamination data.

Line 22: nine environmental pathways, not nine-environmental pathways

Response: corrected throughout the manuscript (see changes in Line 29)

Line 23: Does the SaniPath tool need at trademark?

Response: Thank you for your suggestion. We put trademark “SaniPath™” on first mention the manuscript. See changed on page 1, line 30 and page 2, line 70

Line 66: Please expand a bit on the SaniPath tool

Response: We have added more information on lines 70-74 to describe the SaniPath tool. We also add a citation of Raj et al 2020 tool paper and a link to the SaniPath website.

“The SaniPathTM Exposure Assessment Tool has been implemented in several cities in South Asia and Sub-Saharan Africa (Wang, et al 2017), and includes customized primary data collection on fecal contamination of the residential environment and human exposure behavior and automated Bayesian analyses to estimate risk of fecal exposure [18, 19].”

Line 93: Can you list the date of this meeting?

Response: Yes, we have updated the manuscript to include the date (February 14, 2017). See the date on page 3, line 107.

Data collection: Did you have any controls for the microbial testing?

Response: We did not include a positive control, but we included both a negative control and a field blank. Negative control being a PBS/sterile water that was processed using the same IDEXX methods in the lab and a field blank, meaning PBS/sterile water that was taken to the field and "collected" using the same techniques, to ensure proper sample collection techniques.

See changes on page 5, lines160-164.

Line 164: Please remove "like the mean".

Response: Removed (Line 238)

Define JAGS

Response: JAGS refers to “Just Another Gibbs Sampler”. We have added this in the manuscript on page 8, line 240.

Line 169: Please include references for the "existing literature and formative study data". Is there a table describing this data?

Response: Thank you for your careful review. We have added relevant references. A table describing this data is included in the appendix of the methods paper for the SaniPath Tool by Raj et al, S2. Appendix https://doi.org/10.1371/journal.pone.0234364.s002. (Changes in line 244 and lines 270-277)

[18]. Raj SJ, Wang Y, Yakubu H, Robb K, Siesel C, Green J, et al. The SaniPath Exposure Assessment Tool: A quantitative approach for assessing exposure to fecal contamination through multiple pathways in low resource urban settlements. PLoS One. 2020;15(6):e0234364.

[24]. Wang Y, Mairinger W, Raj SJ, Yakubu H, Siesel C, Green J, et al. Quantitative assessment of exposure to fecal contamination in urban environment across nine cities in low-income and lower-middle-income countries and a city in the United States. Science of The Total Environment. 2022;806:151273. doi: https://doi.org/10.1016/j.scitotenv.2021.151273.

[30]. Amin N, Rahman M, Raj S, Ali S, Green J, Das S, et al. Quantitative assessment of fecal contamination in multiple environmental sample types in urban communities in Dhaka, Bangladesh using SaniPath microbial approach. PLoS One. 2019;14(12):e0221193.

You also repeat the exposure assessment methodology in lines 169-171. Please remove.

Response: Thanks for your careful review. We have revised the section. See changes on page 8, lines 243-250

The second half of the paragraph is repeated. Please revise.

Response: We have revised the whole section to avoid repetition. See changes in lines 243-260.

Data analysis: additional information is needed here. Are there any equations for dose? More information on the software is needed too.

Response: We have added details on the analysis plan including the tools used for the analysis. The dose we used for the SaniPath tool is in the SaniPath Tool paper S2. Appendix https://doi.org/10.1371/journal.pone.0234364.s002. (See changes in lines 252-260.

There is some information in the footnotes under Table 2, but this needs to be included in the methods section.

Response: We have added more information about the behavior questions to the methods section on lines 293-310. We also added more detailed information in the results sections.

The Results section has several typos and the figures are very blurry and hard to review. I had trouble accurately assessing the results given the quality of the figures.

Response: Thanks for checking, we have checked all typos throughout the manuscripts and improved the quality/resolution of the figures.

Line 412: Please include references for "and studies by others".

Response: References added in line 504.

Fecal contamination of produce and street food in Bangladesh and other LMICs has been reported in numerous SaniPath deployments [21] and studies by others [13, 36, 37].

Line 591: Is there a typo for SFD.

Response: we have checked the spelling and there is no typo. “SFD” is the conventional acronym for “Shit Flow Diagram” Please check here: https://sfd.susana.org/ (Line 701)

The conclusion is very long and much of the information discussed should be included in the discussion. Please revise.

Response: Thank you for this thoughtful suggestion. We understand your concern regarding the length of the conclusion. However, as the study covers multiple exposure pathways and neighborhood contexts, we aimed to provide distinct and practical recommendations to inform future interventions and policy actions.

We have carefully reviewed and revised the conclusion to make it more concise and focused, while still retaining the key messages that emerged from our findings. Given the complexity and breadth of the study, a certain level of detail remains necessary to ensure clarity and utility for public health practitioners and decision-makers. We hope the revised version strikes a better balance between brevity and completeness. Thank you again for your feedback.

Reviewer #2:

Overall comments:

The authors investigated fecal exposure from different routes in urban Dhaka using a SaniPath methodology. There are some places where the manuscri

---

## [Decision Letter · Decision Letter 1]

16 Sep 2025

Dear Dr. Amin,

Thank you for submitting your manuscript to PLOS ONE. After careful consideration, we feel that it has merit but does not fully meet PLOS ONE’s publication criteria as it currently stands. Therefore, we invite you to submit a revised version of the manuscript that addresses the points raised during the review process.

We look forward to receiving your revised manuscript.

Kind regards,

D. Daniel, Ph.D.

Academic Editor

PLOS ONE

Journal Requirements:

Reviewers' comments:

Reviewer's Responses to Questions

**Comments to the Author**

Reviewer #2: (No Response)

Reviewer #3: (No Response)

2. Is the manuscript technically sound, and do the data support the conclusions?

Reviewer #2: Yes

Reviewer #3: Partly

3. Has the statistical analysis been performed appropriately and rigorously?

Reviewer #2: I Don't Know

Reviewer #3: I Don't Know

4. Have the authors made all data underlying the findings in their manuscript fully available?

Reviewer #2: Yes

Reviewer #3: No

5. Is the manuscript presented in an intelligible fashion and written in standard English?

Reviewer #2: Yes

Reviewer #3: Yes

Reviewer #2: The authors have adequately addressed most of my previous comments. I just have two comments remaining:

1) I remain confused about the intention of the different behavioral surveys mentioned on page 5. For example, I do not understand why household surveys were conducted with mothers of children <5 years when children <5 were not being assessed for exposure. If these surveys were used for estimating exposure of adults and children aged 5-12 years, why were households with children 5-12 years not targeted instead? And why did school surveys primarily focus on children aged 10-12 instead of children aged 5-12? More largely, it is not clear what information was obtained from these two surveys and how that data was used in combination with the community surveys to estimate exposure.

2) Please carefully check the footnote edits and associated labels that have been changed for Table 2. There is a c in front of the Household surveys label that appears to be an error, and it also looks like the footnote for municipal and non-municipal water is no longer labeled after the relevant text in the table.

Reviewer #3: RE: Most of the analysis methods for this paper were completed by the SaniPath Exposure Assessment tool, whose methods were described in another paper. As a result, this paper is not easy to understand without reading the other paper. I suggest the authors BRIEFLY describe the key inputs and data processing steps of the tool.

Response: This comment is consistent with the first comment from Reviewer 1, which we have addressed in detail. We kindly request the reviewer to refer to our response to that section for further clarification.

Reviewer 3: It is difficult to determine of Reviewer 1’s comments and your response is responsive to this suggestion. Please clarify. Note: The information provided on this website: https://www.sanipath.net/sanipath-approach

Is the type of brief summary I was looking for. A figure/process diagram would also be helpful to show the inputs, processes and outputs of “the tool.” All three are necessary to understand what the tool is and how the collected data is used.

RE: Line 315. Could you provide more details about how the doses were estimated? I presume dose was a combination of a level of e.coli and a model that incorporates differences in ingestion rates and other behaviors. Given that the reader does not have access to the underlying inputs of e.coli and behavior, the reader cannot ascertain the extent to which the differences in dose were attributable to the level of e.coli or the behavior. Please provide a sensitivity analysis for thiss.

Response: Thank you for your comments. We do not think a sensitivity analysis is necessary for this study. We have conducted a series of validation studies for the SaniPath tool as described in Raj et al. 2020.The dose assumptions we used for the SaniPath tool are in the SaniPath Tool paper S2. Appendix https://doi.org/10.1371/journal.pone.0234364.s002. (also see revised analysis description on page 9)

Validation and sensitivity analyses are different things – and the authors have not provided a defensible rationale for why the sensitivity analysis is not necessary (i.e., conducting a validation study is not appropriate). A validation study seeks confirms that the tool is accurately and reliably measuring what it is purported to measure. A sensitivity analysis assesses the relative uncertainty and assesses how changes in each input variable impact the output. Please address.

**Do you want your identity to be public for this peer review?** For information about this choice, including consent withdrawal, please see our Privacy Policy

Reviewer #2: No

Reviewer #3: No

---

## [Author Response · Author response to Decision Letter 2]

11 Nov 2025

Reviewer #2:

The authors have adequately addressed most of my previous comments. I just have two comments remaining:

1) I remain confused about the intention of the different behavioral surveys mentioned on page 5. For example, I do not understand why household surveys were conducted with mothers of children <5 years when children <5 were not being assessed for exposure. If these surveys were used for estimating exposure of adults and children aged 5-12 years, why were households with children 5-12 years not targeted instead? And why did school surveys primarily focus on children aged 10-12 instead of children aged 5-12? More largely, it is not clear what information was obtained from these two surveys and how that data was used in combination with the community surveys to estimate exposure.

Response: Thank you for the questions. Household surveys were conducted and analyzed for children ages 5–12, not for children under 5, and we have corrected this error. School surveys were administered to school-age children (ages 10–12), as this falls within the 5–12 age range and is appropriate for a classroom setting. Older children are also better able to understand the questions and select accurate responses. All surveys were standardized, and behavioral data were combined for the exposure assessment (i.e., adult data from all three survey types were aggregated, and child data from all three survey types were aggregated). Because the SaniPath assessment is designed to be a relatively rapid data collection exercise, providing three survey options—household, school, and community—allows users to either collect more data overall or select the survey types that best match their capacity. Collecting multiple survey types also helps overcome the limitations of accessing specific populations (e.g., high-income households may be harder to reach through household surveys). Using data from all three surveys ultimately increases the amount of information available to estimate exposure. Please find the changes on Page 6, Lines 168-178)

2) Please carefully check the footnote edits and associated labels that have been changed for Table 2. There is a c in front of the Household surveys label that appears to be an error, and it also looks like the footnote for municipal and non-municipal water is no longer labeled after the relevant text in the table.

Response: Thank you for your careful review. We have checked all footnotes and corrected accordingly.

Reviewer #3:

Most of the analysis methods for this paper were completed by the SaniPath Exposure Assessment tool, whose methods were described in another paper. As a result, this paper is not easy to understand without reading the other paper. I suggest the authors BRIEFLY describe the key inputs and data processing steps of the tool.

Response: This comment is consistent with the first comment from Reviewer 1, which we have addressed in detail. We kindly request the reviewer to refer to our response to that section for further clarification.

It is difficult to determine of Reviewer 1’s comments and your response is responsive to this suggestion. Please clarify. Note: The information provided on this website: https://www.sanipath.net/sanipath-approach

Is the type of brief summary I was looking for. A figure/process diagram would also be helpful to show the inputs, processes and outputs of “the tool.” All three are necessary to understand what the tool is and how the collected data is used.

Response: Thank you for your further comment. We have provided a lot of writing to BRIEFLY describe the analysis method in this paper (see line 261 and figure 1). We have added a diagram (Fig 1) and revised the method to incorporate this diagram.

Line 315. Could you provide more details about how the doses were estimated? I presume dose was a combination of a level of e.coli and a model that incorporates differences in ingestion rates and other behaviors. Given that the reader does not have access to the underlying inputs of e.coli and behavior, the reader cannot ascertain the extent to which the differences in dose were attributable to the level of e.coli or the behavior. Please provide a sensitivity analysis for thiss.

Response: Thank you for your comments. We do not think a sensitivity analysis is necessary for this study. We have conducted a series of validation studies for the SaniPath tool as described in Raj et al. 2020.The dose assumptions we used for the SaniPath tool are in the SaniPath Tool paper S2. Appendix https://doi.org/10.1371/journal.pone.0234364.s002. (also see revised analysis description on page 9)

Validation and sensitivity analyses are different things – and the authors have not provided a defensible rationale for why the sensitivity analysis is not necessary (i.e., conducting a validation study is not appropriate). A validation study seeks confirms that the tool is accurately and reliably measuring what it is purported to measure. A sensitivity analysis assesses the relative uncertainty and assesses how changes in each input variable impact the output. Please address.

Response: As to sensitivity analysis, we assume this is not the objective for this paper to do that. The objective of this paper is to use an established analtytical method to conduct exposure assessment in Dhaka not to justify or develop the SaniPath exposure assessment method.

The SaniPath tool uses relatively naive assumptions for intake volume (constant volume or percent based on best literature we can find). Please see the table below:

Table 1. Assumptions and factors used in the SaniPath Tool exposure model

1. U.S. EPA. Exposure Factors Handbook 2011 Edition (Final Report). 2011.

2. SaniPath. SaniPath Phase 1 Data.

3. U.S. EPA. Descriptive Statistics from a Detailed Analysis of the National Human Activity Pattern Survey (NHAPS) Responses. Washington, DC; 1996. doi:EPA/600/R-96/148

4. Dorevitch S, Panthi S, Huang Y, et al. Water ingestion during water recreation. Water Res. 2011;45(5):2020-2028. doi:10.1016/j.watres.2010.12.006

5. Dufour AP, Evans O, Behymer TD, Cantú R. Water ingestion during swimming activities in a pool: a pilot study. J Water Health. 2006;4(4):425-430. http://www.ncbi.nlm.nih.gov/pubmed/17176813. Accessed October 16, 2019.

6. Percent consumed estimate is based on serving size of produce or street food that an adult could be expected to eat. For child estimates, the serving size is estimated to be half that of the adult portion.

In this table, you can see the "multiplication factor" between adults and children are within a difference of 1 log10 scale, except flood water and drain water (within a log10 scale difference of 1.22). Therefore, the difference in exposure between adults and children could be attributed to the intake assumption up to around 1 log10 difference (not a huge impact). For comparing different pathways, the largest difference of multiplication factor is around 4.2 log10 difference (drinking water vs. drain water). Although the number seems to be large, the idea that a person may drink 10,000 times more volume of drinking water compared to drain water seems to be reasonable. We agree that a sensitivity analysis can be helpful in a Monte Carlo simulation to examine the impact of different parameters and assumptions on the results. Since all the intake volumes are constant multiplication factors, making a linear relationship with exposure (e.g., making intake volume 10 times larger will lead to 10 times larger exposure). Given the very limited information (especially in variation) about intake volume available, we are not sure how useful the results of sensitivity analysis.

---

## [Decision Letter · Decision Letter 2]

7 Dec 2025

Variation in exposure in neighborhoods of Dhaka, Bangladesh across different environmental pathways: The influence of human behavior on fecal exposure in urban environments

PONE-D-25-07134R2

Dear Dr. Amin,

We’re pleased to inform you that your manuscript has been judged scientifically suitable for publication and will be formally accepted for publication once it meets all outstanding technical requirements.

Kind regards,

D. Daniel, Ph.D.

Academic Editor

PLOS One

Additional Editor Comments (optional):

Reviewers' comments:

Reviewer's Responses to Questions

**Comments to the Author**

Reviewer #2: All comments have been addressed

2. Is the manuscript technically sound, and do the data support the conclusions?

Reviewer #2: Yes

3. Has the statistical analysis been performed appropriately and rigorously?

Reviewer #2: I Don't Know

4. Have the authors made all data underlying the findings in their manuscript fully available?

Reviewer #2: Yes

5. Is the manuscript presented in an intelligible fashion and written in standard English?

Reviewer #2: Yes

Reviewer #2: The authors have adequately addressed my previous comments. One of the minor errors I pointed out related to footnote labels in Table 2 remains in the revised text (where there is a superscript c that does not correspond to any footnote in front of the Household surveys sub-heading within the table, but this could be corrected during copy editing.

**Do you want your identity to be public for this peer review?** For information about this choice, including consent withdrawal, please see our Privacy Policy

Reviewer #2: No

---

## [Editor Report · Acceptance letter]

PONE-D-25-07134R2

PLOS One

Dear Dr. Amin,

I'm pleased to inform you that your manuscript has been deemed suitable for publication in PLOS One. Congratulations! Your manuscript is now being handed over to our production team.

Kind regards,

on behalf of

Dr. D. Daniel

Academic Editor

PLOS One